# Successful Treatment of *Staphylococcus aureus* Prosthetic Joint Infection with Bacteriophage Therapy

**DOI:** 10.3390/v13061182

**Published:** 2021-06-21

**Authors:** Claudia Ramirez-Sanchez, Francis Gonzales, Maureen Buckley, Biswajit Biswas, Matthew Henry, Michael V. Deschenes, Bri’Anna Horne, Joseph Fackler, Michael J. Brownstein, Robert T. Schooley, Saima Aslam

**Affiliations:** 1Department of Infectious Diseases and Global Public Health, University of California San Diego, La Jolla, CA 92093, USA; ccramirezsanchez@health.ucsd.edu (C.R.-S.); rschooley@health.ucsd.edu (R.T.S.); 2Department of Surgery, University of California San Diego, La Jolla, CA 92093, USA; fbgonzales@health.ucsd.edu (F.G.); mabuckley@health.ucsd.edu (M.B.); 3Biological Defense Research Directorate, Naval Medical Research Center, Fort Detrick, MD 21702, USA; Biswajit.biswas.civ@mail.mil (B.B.); matthew.s.henry23.ctr@mail.mil (M.H.); michael.v.deschenes.ctr@mail.mil (M.V.D.); 4The Geneva Foundation, Tacoma, WA 98402, USA; 5Leidos, Reston, VA 20190, USA; 6Adaptive Phage Therapeutics, Gaithersburg, MD 20878, USA; bhorne@aphage.com (B.H.); jfackler@aphage.com (J.F.); mjbrownstein@gmail.com (M.J.B.); 7Center for Innovative Phage Therapy and Applications, University of California San Diego, La Jolla, CA 92093, USA

**Keywords:** bacteriophage, phage therapy, prosthetic joint

## Abstract

Successful joint replacement is a life-enhancing procedure with significant growth in the past decade. Prosthetic joint infection occurs rarely; it is a biofilm-based infection that is poorly responsive to antibiotic alone. Recent interest in bacteriophage therapy has made it possible to treat some biofilm-based infections, as well as those caused by multidrug-resistant pathogens, successfully when conventional antibiotic therapy has failed. Here, we describe the case of a 61-year-old woman who was successfully treated after a second cycle of bacteriophage therapy administered at the time of a two-stage exchange procedure for a persistent methicillin-sensitive *Staphylococcus aureus* (MSSA) prosthetic knee-joint infection. We highlight the safety and efficacy of both intravenous and intra-articular infusions of bacteriophage therapy, a successful outcome with a single lytic phage, and the development of serum neutralization with prolonged treatment.

## 1. Introduction

Successful joint replacements provide improved quality of life for millions of patients each year with pain relief and improved mobility [1,2]. However, in a minority of cases, the prosthetic joint becomes infected, estimated to be 6.9/1000 patients in the U.S. for hip and knee prostheses [3]. This is associated with significant decline in quality of life as well as increased cost to the healthcare system [4,5]. In an ageing population, the number of joint replacements continues to increase, and thus we expect to see increasing numbers of prosthetic joint infections (PJI). In the setting of device placement, such as for prosthetic joints, bacterial infections largely arise from biofilms, which make the infectious process more difficult to resolve and are usually incurable without device replacement [3,6]. 

Biofilms are a conglomerate of bacteria that are contained in an extracellular polymeric matrix formed by proteins, exopolysaccharides, and small molecules that are adherent to an underlying surface [7]. This matrix confers protection to the microorganisms from host defenses and results in up to a 1000-fold increase in the minimum inhibitory concentration (MIC) to antibiotics, making biofilm-based infections very difficult to resolve without removal of the infected device [6,8,9]. 

The use of bacteriophages emerged in the early 20th century and has now resurfaced as a treatment option for multidrug-resistant bacteria as well as biofilm-based infections [10]. Bacteriophages, or phages, are ubiquitous viruses that require bacterial hosts to survive. There are two types of phages, temperate and lytic. Temperate phages infect their bacterial host and integrate their genetic material into the bacterial genome for proliferation. Lytic phages use the bacterial replication equipment to create phage progeny, which lyse the bacterial cell wall to release new phages and continue the cycle of infecting and lysing bacteria [10,11]. There are several recent reports of successful outcomes using bacteriophage therapy (BT) for prosthetic and cardiac device infections [8,9,12,13,14].

We present the case of a 61-year-old woman who was treated with two cycles of lytic BT to eradicate a methicillin-sensitive *Staphylococcus aureus* (MSSA) right knee PJI that had persisted for several years. Top-line results from the experience of BT in this patient were previously reported as part of a case series of the first 10 cases of BT treated at our institution [12].

## 2. Materials and Methods

### 2.1. Bacteriophage Susceptibility Testing

The patient’s MSSA isolates were tested in vitro for susceptibility to anti-staphylococcal phages prior to BT initiation. In vitro susceptibility testing was performed via the double agar overlay method or Biolog method [12,15]. Briefly, the agar overlay method consists of adding serial dilutions of phage to a lawn of bacterial isolate grown on an agar plate and observing individual plaques that form if the isolate is susceptible [12]. The Biolog method relies on a standardized bacterial suspension incubated with several dilutions of phage in a 96-well plate. Bacterial respiration is measured to assess bacterial growth [12,16].

### 2.2. Bacteriophages Used

For the first treatment cycle, a cocktail of three phages was used, AB-SA01 (Ampliphi Biosciences, now Armata Pharmaceuticals, Marina Del Rey, CA, U.S.). The AB-SA01 cocktail included phage J-Sa36 (5.3 × 10^8^ plaque forming units, PFU/mL), phage Sa83 (7.7 × 10^8^ PFU/mL) and phage Sa87 (5.9 × 10^8^ PFU/mL); this combination has previously reported anti-biofilm activity [7]. The patient received one intra-articular dose followed by intravenous (IV) infusions every 12 h for 2 weeks. The endotoxin level of the cocktail was <250 EU/mL (<5EU/kg per dose as required by the FDA) [12]. Cefazolin (2 g IV every 8 h, 3 doses per day) was given for 6 weeks. Each phage dose was 2 h before or after an antibiotic dose. 

For the second treatment cycle, a single lytic phage, SaGR51ø1 (Adaptive Phage Therapeutics, Gaithersburg, MD, U.S.) was used at a concentration of 2.89 × 10^10^ PFU/mL. The calculated endotoxin level was <1 EU/mL. The patient received a single intraoperative dose of phage as well as IV treatments every 12 h for 6 weeks. In addition to BT, the patient received concomitant IV cefazolin (2 g every 8 h) for 6 weeks. 

### 2.3. Serum Neutralization Assay 

We conducted a serum neutralization assay of phage prior to the second course of BT by incubating 2.4 × 10^6^ PFU of phage SaGR51ø1 with 1:100 dilution of patient serum and then sampling at different time points. We then counted PFU for each sample using the agar overlay method described earlier. Only serum neutralization to SaGR51ø1 was performed.

### 2.4. Clinical Treatment Protocol 

The patient commenced BT after informed consent and approval from the FDA and local institutional review board for a single patient use investigational new drug application. Local biohazard use authorization was obtained as well. We collected weekly serum samples at baseline prior to BT initiation and then each week while the patient was on therapy for both BT cycles. We also collected weekly blood cultures (processed at the clinical microbiology laboratory, Center for Advanced Laboratory Medicine, University of California San Diego (UCSD)) and the following clinical laboratory data-complete blood count, complete metabolic panel, sedimentation rate (ESR), and C-reactive protein (CRP) levels in each cycle. During the first BT course, the patient underwent baseline and then weekly arthrocentesis as well, and we collected samples for synovial cell counts and cultures. During the second treatment cycle, baseline synovial cultures were obtained during surgery followed by arthrocentesis for cell count and bacterial culture at the end of BT. 

### 2.5. Surgical Treatment Protocol

After recurrence of the infection following the first round of BT, the patient underwent a right-knee resection arthroplasty, including removal of all implants, wash-out of the visibly infected joint space, and placement of a stemmed articulating vancomycin/tobramycin cement spacer and biodegradable antibiotic beads. She underwent excision of draining sinuses with complex wound closure. After closure of the arthrotomy, 10 mL of bacteriophage solution (SaGR51ø1) was injected into the joint cavity using the sterile technique. Immediately after successful re-approximation of the wound, a negative pressure wound device was placed over the fascial and skin closure. 

## 3. Results

### 3.1. Clinical History Prior to Phage Therapy

The patient is a 61-year-old woman suffering from osteoarthritis who underwent a right total knee arthroplasty (TKA) in 1999 followed by revision of the joint in 2016 (timeline in Figure 1). Soon after the revision, she developed right knee swelling, fever, and chills. Synovial fluid cultures at the time grew *S. aureus* (reportedly MSSA but microbiology results were not available to us) at another institution, and she was treated with several weeks of IV vancomycin. Since then, she had recurrent swelling and pain in her right knee requiring multiple surgical debridement and washouts, and then finally a two-stage TKA exchange in 2018. This surgery was complicated by a recurrent MSSA infection manifested by knee swelling, pain, and draining sinus tracts and leading to limited mobilization due to severe chronic pain. Synovial fluid and drainage cultures grew MSSA, and the infection persisted despite several courses of IV cefazolin and oral suppressive antibiotics. Due to multiple surgeries, an above-knee amputation was recommended as a last-resort procedure. She then came to our institution seeking a second opinion as well as consideration for BT. 

When the patient first visited UCSD, she was on IV cefazolin and oral rifampin and amoxicillin. She had ongoing swelling, pain, and limited range of motion and weight-bearing of her right knee as well as a draining sinus tract. She underwent an arthrocentesis in February 2019 that grew MSSA. 

#### 3.1.1. First Course of BT

The patient was started on AB-SA01 on 12 March 2019. The initial course consisted of a one-time intra-articular knee injection of AB-SA01 followed by IV infusions ever 12 h for 2 weeks in conjunction with IV cefazolin (every 8 h) for a 6-week period (1 March–11 April 2019). There were no adverse effects related to BT. The patient’s weekly laboratory tests, including liver function tests, renal function and complete cell blood count, remained stable. Her inflammatory markers, CRP and ESR, remained elevated at the end of the 2 weeks of BT: CRP from 2.9 mg/dL to 4.2 mg/dL and ESR from 84 mm/h to 83 mm/h. Clinically, she felt better with resolution of drainage from the sinus tract and reduction in pain. Our initial plan had been to treat her for 6 weeks with both phage and antibiotics. However, BT stopped at the 2-week mark because the manufacturer was unable to supply more product for us to use. Blood and synovial fluid cultures during the first course of BT remained negative. 

By 5 days following discontinuation of cefazolin (after completion of 6 weeks of antibiotic therapy), the patient developed worsening knee pain, swelling, and recurrent drainage from the knee; drainage culture grew MSSA again (Figure 1). The antimicrobial susceptibility profile (resistance to rifampin and penicillin only) was identical to the MSSA isolates obtained prior to the start of BT. We were unable to test for resistance to AB-SA01 as the phage cocktail was no longer available to us. The patient was then treated with 8 weeks of cefazolin ending on 1 August 2019 and then restarted on 23 August 2019 due to clinical worsening and recurrent positive synovial fluid culture with MSSA (same antimicrobial susceptibility pattern as earlier) while on oral suppression with trimethoprim-sulfamethoxazole in the interim. 

#### 3.1.2. Second Course of BT

The patient underwent a two-stage repair on 25 September 2019 with removal of infected hardware and spacer implantation (Figure 2). She received a one-time intra-articular dose of 10 mL of SaGR51ø1 (Figure 3) on 25 September 2019 and daily infusions every 12 h for 6 weeks starting 26 September 2019 in conjunction with IV cefazolin (every 8 h for 6 weeks). All 14 intra-operative samples sent for culture on 25 September 2019 grew MSSA, and they were susceptible to the phage (one culture grew *Candida albicans* in broth only, which was considered as a possible contaminant but treated with oral fluconazole). The antimicrobial susceptibility profile of MSSA (resistance to rifampin and penicillin only) was identical to previous MSSA isolates. Synovial fluid, sampled towards the end of BT on 1 November 2019, was non-inflammatory with only 262 WBC/mL (12% PMNs) and negative bacterial culture. No adverse events related to BT were noted. The patient’s weekly labs remained stable. Her CRP and ESR showed improvement from 9.23 mg/dL to 1.43 mg/dL, and from 85 mm/h to 63 mm/h, respectively. Since then, the patient has undergone multiple (n = 25) synovial fluid and wound cultures that have been negative for MSSA. The last culture was on 9 December 2020.

### 3.2. Results of Serum Neutralization

We assessed serum neutralization of SaGR51ø1 at the initiation (Day1) and end (Day 43) of the 2nd phage treatment cycle, as noted in Figure 4. Serum neutralization activity was significantly increased by the end of BT. Notably, phage concentration decreased in the presence of serum obtained on Day 1 of treatment (prior to phage administration), and on Day 43 (post treatment). The decrease was detected in the first hour of testing. No inhibition was seen in phosphate-buffered saline (PBS) control where viable phages remained detectable. The results show that neutralizing antibodies were present in the patient’s serum at baseline with an increased concentration by the end of treatment. Prolonged exposure resulted in phage reduction in the PBS controls but to a lesser extent than serum samples at all time points.

## 4. Discussion

We demonstrated a successful outcome in a patient with recalcitrant *S. aureus* PJI when prolonged BT was used in conjunction with surgery and antibiotics. This success occurred in the setting of failure of a previous two-stage TKA and prolonged antibiotic courses, as well as a short course of BT without surgery. 

Phage therapy has been given on a compassionate-use basis in a variety of clinical settings—both for the treatment of multidrug-resistant bacterial infections and also for biofilm-based infections. In our case, the patient had a very recalcitrant infection caused by a drug-susceptible organism. Systemic antibiotics are generally effective against planktonic organisms but are frequently ineffective against the same organisms when they are embedded within a biofilm matrix [9] for a variety of reasons. These include biofilm penetration, hypoxic micro-environment, presence of sessile cells rather than actively dividing one, and presence of persister cells [6,17]. In the case of staphylococci, their ability to produce an adherent multilayered biofilm on implanted biomedical devices is considered a major virulence factor, and in our case, made the treatment challenging despite the antibiotic susceptibilities of the isolate [18]. The ability of some phages to cause biofilm disruption [7,8,19] makes them attractive for the treatment of device infections. 

Our patient failed the first BT course after receiving an intra-articular injection of three phages followed by IV dosing for 2 weeks. It is unclear if failure was related to disease burden, i.e., lack of a drainage procedure (arthroscopic/surgical washout, surgical joint revision, and/or percutaneous drain placement), phage concentration, and/or duration of phage treatment. In the absence of drainage, there may have been high numbers of *S. aureus* and low multiplicity of infection (MOI), i.e., the ratio of phage to bacteria. The relatively low concentrations of phages in the cocktail that was originally used may have contributed to this. Unfortunately, it is difficult to estimate MOI when one treats a biofilm [20,21]. Perhaps a longer treatment period in the first round of BT would have helped [22]. Even though intra-articular and IV phage was administered in both BT courses, the second course also included hardware removal and physical washout of the infected joint cavity leading to a reduction in organism and biofilm burden; this most likely played a role in the resolution infection. Previous cases of BT for PJI include long IV courses of phage as well as local joint injection along with drainage of the infected joint (via arthroscopy, local drain placement and/or surgical debridement) [8,14,23,24,25]. Additionally, the concentration of phage was different in both cycles as well—10^8^ PFU/mL in the first and 10^10^ PFU/mL in the second; this higher concentration may have played a role in the successful outcome, as seen in another successful case in which 6.3 × 10^10^ PFU/mL was used [8]. The appropriate dose, duration, and route of administration of phage in the treatment of PJI is unclear and may be best addressed by clinical trials.

Phage literature in general supports use of multiple phages in combination. This is based on the idea that multiple phages may have an expanded spectrum of activity and reduction of phage resistance emerging during treatment [19,26]. Given the density of biofilm matrix, the low metabolic activity, and development of phage resistance [26], it is argued that phage cocktails rather than monotherapy may be more effective in treating infections [19]. However, this report supports the use of a single *S. aureus* phage in the setting of personalized treatment where expansion of the host range is not an issue. Many *S. aureus* phages used for therapy are related to the well-known phage K [27] and use a similar bacterial receptor binding protein for host cell attachment [28]. Thus, in the setting of *S. aureus* infections, we think that a cocktail of phages might not provide additional benefit compared to a single phage. In our case, the use of phage monotherapy was successful in the second BT cycle despite the presence of neutralization related to antibody formation. We noted slight regrowth in vitro of our clinical isolate after about 20 h as shown in Figure 3; however, we determined that this uptick does not necessarily represent clinically relevant phage-resistant bacteria, which was confirmed by the success in treatment of our isolate and the negative cultures after treatment. 

It is not surprising that phages stimulate an adaptive immune response, which could potentially affect BT [25,29]. One mouse model study noted development of humoral activity against *S. aureus* phages delivered orally [30]. The authors also noted that intraperitoneal injection of phage led to higher titers of neutralizing sera compared to oral ingestion. A human study of 122 patients treated with BT found that in vitro antibody neutralization following BT may depend on the route of phage administration as well; local delivery (into abscess cavities or fistulae, inhalation) may induce a higher humoral response than oral delivery [31]. Intraperitoneal or IV therapy was not evaluated in that study. In our patient, it is possible that the first BT course led to an immune response that was cross-reactive with the phage used in the second BT course. However, successful clinical outcome was achieved despite the presence of in vitro serum neutralization at the start of the second course, which is very encouraging. Direct intra-articular phage injection could potentially have played a role in avoiding serum neutralization as well, especially after joint wash-out at the time of surgery. 

Lastly, we demonstrated the safety and ease of administration of BT. The patient did not have any adverse event related to either cycle of BT. We taught the patient how to self-administer BT with the first in-person dose and provided detailed storage and step-by-step administration instructions. She was able to self-administer BT as an outpatient through an indwelling peripherally inserted central catheter without any problems or missed doses. 

We would like to highlight several points about our case: (1) we achieved a successful outcome with phage monotherapy in the second BT course, (2) the outcome was successful despite the presence of pretreatment antibodies leading to in vitro serum neutralization in the second BT course, (3) safety of phage therapy and lack of adverse events was observed in both BT courses when given IV as well as intra-articular, and (4) the treatment response has been durable as the patient is now 20 months past the second BT course with no recurrence of MSSA.

## 5. Conclusions

We describe our successful experience using BT as an adjunct to existing standard of care consisting of surgery and systemic antibiotics for the resolution of a recalcitrant MSSA PJI. Further research into optimal dose and route of phage administration and the influence of phage immune response to a successful clinical outcome is needed.

## Figures and Tables

**Figure 1 viruses-13-01182-f001:**
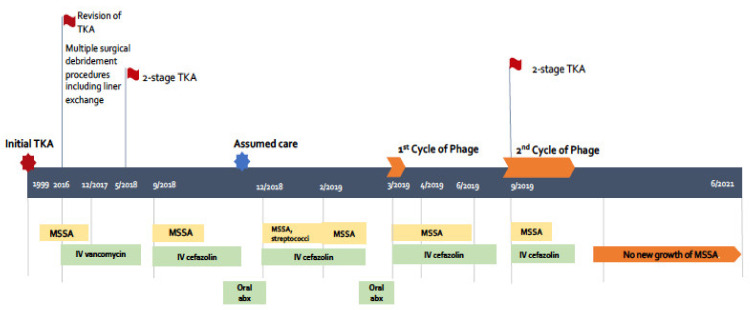
Timeline schematic of patient’s care (not to scale). TKA: total knee arthroplasty, MSSA: methicillin sensitive *Staphylococcus aureus*, IV: intravenous, abx: antibiotic.

**Figure 2 viruses-13-01182-f002:**
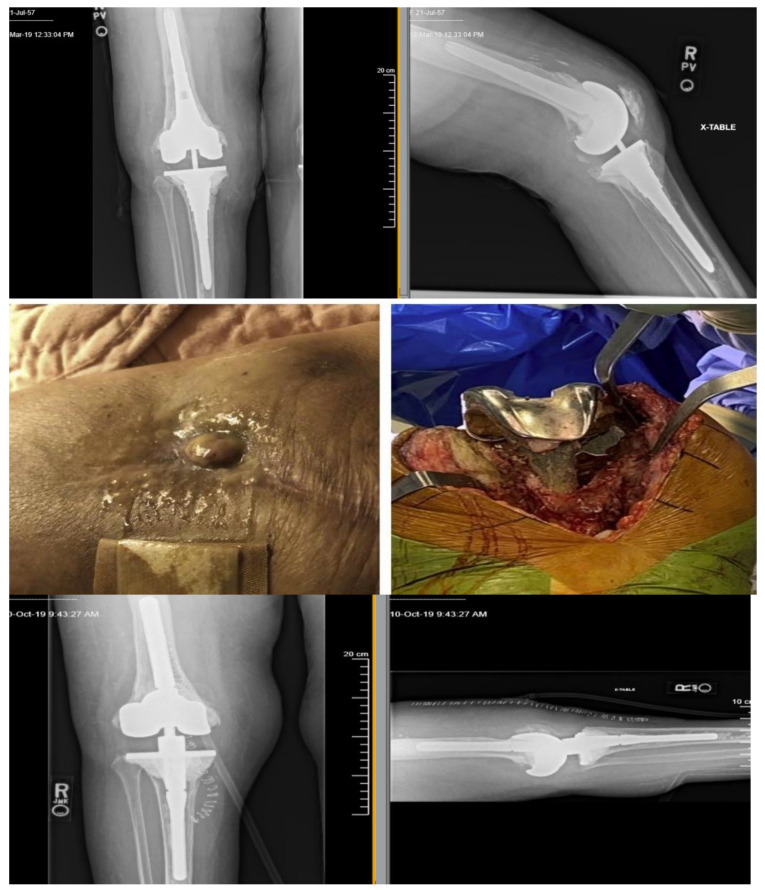
Top row: X-rays of infected right prosthetic joint showing a cortical lucency around the medial aspect of the tibial stem on the frontal view and the anterior aspect of the distal femoral stem on the lateral view. Middle row: Draining sinus tract when infection recurred after completion of first course of phage therapy (**left**) and intraoperative view of the right knee and prosthesis during 2-stage repair with removal of infected hardware and spacer implantation (**right**). Bottom row: X-rays of the new prosthesis at which time phage was instilled into the joint space for second course of phage therapy.

**Figure 3 viruses-13-01182-f003:**
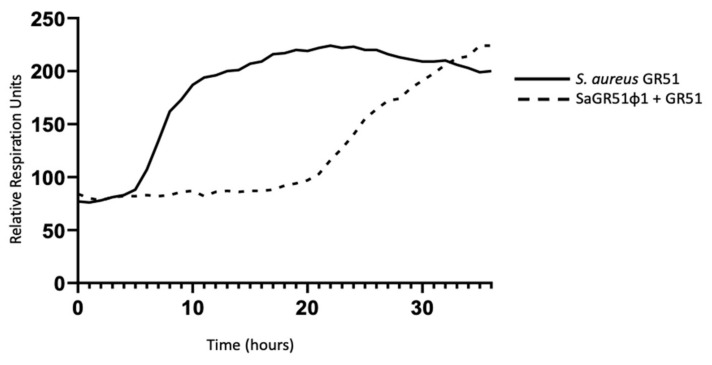
Time-kill curve using the Biolog assay demonstrating in vitro lysis of the patient’s bacterial isolate (GR51) by phage SaGR51ø1. This was conducted in a micro-well plate, and colonies were not recovered and tested for resistance to the phage.

**Figure 4 viruses-13-01182-f004:**
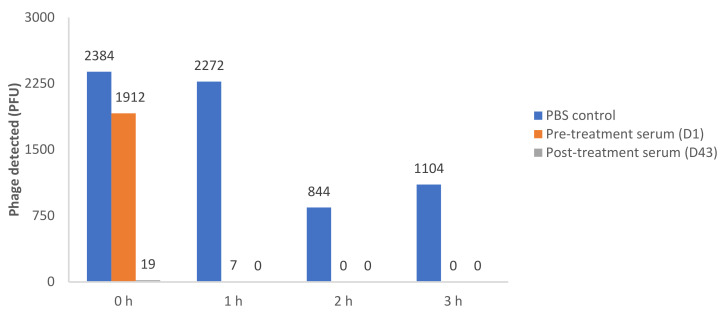
Recovery of viable phage after incubation with patient’s serum and 2.4 × 10^6^ PFU of SaGR51ø1 phage. Phage concentration in the presence and absence of serum at 4 different time points (0, 1, 2 and 3 h). PBS: Phosphate buffered saline; D1: Day 1; D43: Day 43.

## Data Availability

The data presented in this study are available on request from the corresponding author. The data are not publicly available due to patient privacy.

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
