# Peer review of "Successful Treatment of Staphylococcus aureus Prosthetic Joint Infection with Bacteriophage Therapy"

_viruses, 2021, doi:10.3390/v13061182_

Round 1

Reviewer 1 Report

This is an interesting case report about phage therapy in a patient with recurrent prosthetic knee infection due to S. aureus. More details could be discussed:

  • A new 2-stage approach (performed in this patient) by itself would be associated with a significant percentage of success, without the use of phage. Here authors could discuss this point as it is not a "dead-end" clinical situation, but a common clinical situation (it is a little bit surprising for me that the FDA approved phage in this indication). Please add in the abstract that the patient was managed finally with a 2-stage approach.
  • Please provide the X-ray before and after prosthesis exchange.
  • The neutralization of phages by the serum is interesting. Is the synovial fluid could inhibit phages in the same way? it could be discussed, because serum could neutralize phages administered via the intravenous route, but not via the intra articular route, especially as phages are directly in the contact of the biofilm in this way. Please discuss the mechanism of serum inhibition (production of antibodies that neutralized the phages?).

Author Response

First reviewer comments:

Comments and Suggestions for Authors

This is an interesting case report about phage therapy in a patient with recurrent prosthetic knee infection due to S. aureus. More details could be discussed:

A new 2-stage approach (performed in this patient) by itself would be associated with a significant percentage of success, without the use of phage. Here authors could discuss this point as it is not a "dead-end" clinical situation, but a common clinical situation (it is a little bit surprising for me that the FDA approved phage in this indication). Please add in the abstract that the patient was managed finally with a 2-stage approach.

Answer: The patient had previously failed a 2-stage procedure for treatment of staphylococcal PJI as well with several debridement and washout surgeries, which is why we applied for a compassionate use indication. As recommended, we have updated the abstract to add in that the final surgery consisted of a 2-stage procedure.

Please provide the X-ray before and after prosthesis exchange.

Answer: We have added the X-rays as requested.

The neutralization of phages by the serum is interesting. Is the synovial fluid could inhibit phages in the same way? it could be discussed, because serum could neutralize phages administered via the intravenous route, but not via the intra articular route, especially as phages are directly in the contact of the biofilm in this way. Please discuss the mechanism of serum inhibition (production of antibodies that neutralized the phages?).

Answer: We agree it is possible that intra-articular phage administration was important to help avoid serum neutralization. We discuss activation of the humoral immune system in the discussion section. We also added the following statement: “Direct intra-articular phage injection could potentially have played a role in avoiding serum neutralization as well especially after joint wash-out at the time of surgery.”

Reviewer 2 Report

The authors document their experience with the treatment of a TKA PJI due to MSSA in a 61-year-old female patient. This case was previously reported as part of a case series from the same authors (Aslam et al., 2021, Case #7), and it is great to see a more detailed description on this specific case be published. The patient history, course of treatment, applied phages, and clinical and results are well documented. The manuscript is quite good, but I request finer resolution on some aspects to avoid confusion by readers. After these minor revisions for clarity and completeness, I recommend this article for publication and hope that the authors will continue to publish the results of their experiences with phage therapy.

Major comments

  • The authors must state clearly in the article that this patient is also represented as part of their case series that was previously published. Ie., “The experience of phage therapy with this patient was also reported as part of a case series of the first 10 cases of phage therapy treated at our institute”, or something to that degree. This is to avoid patients being counted twice in different publications.

Minor comments

  • Small changes to word order, italicization, abbreviations, hyphenation, and grammar can be found in the accompanying tracked-changes version of the manuscript. Please make the suggested changes.
  • Figure 4 should be “cleaned up” a little prior to publication

Content comments

  • Abstract & start of discussion: the first round of phage therapy was a failure (with probably reasons, but a failure none-the-less), so please reword that the success was after a second, prolonged round of BT, rather than “successfully treated with two rounds of BT”. Suggestions can be found in the track-changes.
  • In consideration of the main audience of Viruses (ie. researchers), it would be better to be clear that PJI is a type of SSI and put more emphasis on PJI (it is in the title and abstract, then SSI is suddenly the focus in the introduction).

M&M / Results

  • Please specify, for the first round of treatment with AB-SA01, if phages were given every 12h (currently written twice daily, whereas during the 2nd round, it is specified every 12h).
    • Also please change/clarify the administration of cefazolin – it is written that it is given with BT, but also that it is given 4x/day
  • Could the authors please elaborate on the difference in reported endotoxin level of the two phage products? Gram+ organisms inherently do not produce endotoxin so it is not clear why AB-SA01 was ~250 EU/mL whereas the single phage was <1 EU/mL.
  • Please specify why serum neutralization to AB-SA01 was not performed? I imagine due to the lack of supply and it is fine that these experiments were not done, but would help to state that these experiments were not performed. Ie. “Only serum neutralization to SaGR51ø1 was performed” would be sufficient to state.
  • Please specify for the surgical treatment protocol, that this was after the recurrence of infection following the first round of BT (suggestion in track changes)
  • Were any analyses performed to ensure that the MSSA isolate was the same from the start of BT until the 2nd course? (ie. AST, RFLP, genome sequencing?)

Discussion

  • Please include some factors for what you mean by “optimal approach” ie. route of administration, concentration, frequency, etc.
  • Please expand upon what is meant by “drainage procedure”? ie. placing a drain, surgical revision, debridement. There are several treatment protocols developing for the treatment of PJI with phages and there is debate between physicians about placing a drain (risk of infection vs prolonged access to the site of infection)
  • Please change “true resistant bacteria” to “clinically-relevant resistant bacteria” (see track changes)
  • Due to the high presence of serum antibodies against phage SaGR51ø1 at the start of the second round of treatment, do the authors speculate that this could be cross reactivity to phages in the AB-SA01, especially given that most S. aureus phages used for therapy are closely-related to phage K? I would suggest that authors look for antibody neutralization in serum taken pre-treatment with AB-SA01 as the text states that baseline serum samples were taken, but not insist as a requirement for publication.
  • General: This discussion would benefit from some reorganization by combining the final paragraph (“The appropriate duration of BT..”) with the earlier paragraph on (“The optimal approach…”), as well as by moving the “We would like to highlight…” to the final paragraph of the discussion.
  • I would also add a time-to-follow up with the number of years/months as part of the 1 of your highlights, as it shows how long-lasting the treatment effect has been, which is great.

Author Response

Second reviewer:

Comments and Suggestions for Authors

The authors document their experience with the treatment of a TKA PJI due to MSSA in a 61-year-old female patient. This case was previously reported as part of a case series from the same authors (Aslam et al., 2021, Case #7), and it is great to see a more detailed description on this specific case be published. The patient history, course of treatment, applied phages, and clinical and results are well documented. The manuscript is quite good, but I request finer resolution on some aspects to avoid confusion by readers. After these minor revisions for clarity and completeness, I recommend this article for publication and hope that the authors will continue to publish the results of their experiences with phage therapy.

Major comments

The authors must state clearly in the article that this patient is also represented as part of their case series that was previously published. Ie., “The experience of phage therapy with this patient was also reported as part of a case series of the first 10 cases of phage therapy treated at our institute”, or something to that degree. This is to avoid patients being counted twice in different publications.

Answer: Thank you for pointing out this omission on our part. The suggested sentence is now included in the revised manuscript.

Minor comments

Small changes to word order, italicization, abbreviations, hyphenation, and grammar can be found in the accompanying tracked-changes version of the manuscript. Please make the suggested changes.

Answer: We appreciate the reviewer comments in the tracked pdf and have accepted them in the revised paper.

Figure 4 should be “cleaned up” a little prior to publication

Answer: Thank you. We have deleted the grid-lines and border lines that were part of the original figure.

Content comments

Abstract & start of discussion: the first round of phage therapy was a failure (with probably reasons, but a failure none-the-less), so please reword that the success was after a second, prolonged round of BT, rather than “successfully treated with two rounds of BT”. Suggestions can be found in the track-changes.

Answer: We completely agree with the reviewer and have revised the language to make this distinction clear. It is because the first round failed that we attempted again with the second phage course.

In consideration of the main audience of Viruses (ie. researchers), it would be better to be clear that PJI is a type of SSI and put more emphasis on PJI (it is in the title and abstract, then SSI is suddenly the focus in the introduction).

Answer: Thank you for pointing this out. We have revised the language in the updated manuscript. We have deleted the term “surgical site infections”.

M&M / Results

Please specify, for the first round of treatment with AB-SA01, if phages were given every 12h (currently written twice daily, whereas during the 2nd round, it is specified every 12h).

Answer: We apologize for this difference in terminology and have corrected it. Both phages were given every 12 hours.

Also please change/clarify the administration of cefazolin – it is written that it is given with BT, but also that it is given 4x/day.

Answer: We have clarified the language to state that cefazolin was given every 8 hours for 3 doses per day; the phage doses were 2 hours before or after the antibiotic dose.

Could the authors please elaborate on the difference in reported endotoxin level of the two phage products? Gram+ organisms inherently do not produce endotoxin so it is not clear why AB-SA01 was ~250 EU/mL whereas the single phage was <1 EU/mL.

Answer: We agree that staphylococci do not produce endotoxin. However, the FDA requires us to formally measure and submit an endotoxin level as part of the eIND application. The different phage products administered to our patient were made by different manufacturers and difference in how the results are reported are due to different detection thresholds of the tests used.

Please specify why serum neutralization to AB-SA01 was not performed? I imagine due to the lack of supply and it is fine that these experiments were not done, but would help to state that these experiments were not performed. Ie. “Only serum neutralization to SaGR51ø1 was performed” would be sufficient to state.

Answer: Unfortunately, Ampliphi Biosciences was bought out by a different company just as we were about to start treatment with AB-SA01. The new company was not interested in working with us. We have updated the revised manuscript with the recommended statement: “Only serum neutralization to SaGR51ø1 was performed”.

Please specify for the surgical treatment protocol, that this was after the recurrence of infection following the first round of BT (suggestion in track changes)

Answer: Thank you for pointing out this omission. This sentence has been revised as suggested in the tracked pdf.

Were any analyses performed to ensure that the MSSA isolate was the same from the start of BT until the 2nd course? (ie. AST, RFLP, genome sequencing?)

Answer: The MSSA isolates from both courses of BT therapy were identical in terms of antimicrobial susceptibility testing (resistance to penicillin and rifampin only) and we did not carry out any additional testing, such as sequencing. We have added this additional information to the revised manuscript.

Discussion

Please include some factors for what you mean by “optimal approach” ie. route of administration, concentration, frequency, etc.

Answer: This exact sentence is now deleted from the revised manuscript. We have replaced with the following: “The appropriate dose, duration, and route of administration of phage in the treatment of PJI is  unclear and may be best addressed by clinical trials.”

Please expand upon what is meant by “drainage procedure”? ie. placing a drain, surgical revision, debridement. There are several treatment protocols developing for the treatment of PJI with phages and there is debate between physicians about placing a drain (risk of infection vs prolonged access to the site of infection)

Answer: We have expanded on this in the revised discussion section, specifically to mention that a drainage procedure may consist of arthroscopic or surgical washout, surgical revision, or percutaneous drain placement.

Please change “true resistant bacteria” to “clinically-relevant resistant bacteria” (see track changes)

Answer: We have revised the statement as suggested by the reviewer.

Due to the high presence of serum antibodies against phage SaGR51ø1 at the start of the second round of treatment, do the authors speculate that this could be cross reactivity to phages in the AB-SA01, especially given that most S. aureus phages used for therapy are closely-related to phage K? I would suggest that authors look for antibody neutralization in serum taken pre-treatment with AB-SA01 as the text states that baseline serum samples were taken, but not insist as a requirement for publication.

Answer: Yes, we agree with the reviewer that it is possible there was cross-reactivity to the phages used in both treatment cycles. As the journal has given us 5 days to submit a revised manuscript, unfortunately we do not have time to test the original baseline serum (i.e. prior to administration of AB-SA01).

General: This discussion would benefit from some reorganization by combining the final paragraph (“The appropriate duration of BT..”) with the earlier paragraph on (“The optimal approach…”), as well as by moving the “We would like to highlight…” to the final paragraph of the discussion.

Answer: Thank you for pointing this out. We have re-organized the discussion section as suggested.

I would also add a time-to-follow up with the number of years/months as part of the 1 of your highlights, as it shows how long-lasting the treatment effect has been, which is great.

Answer: The patient is now 20 months past the second BT course with no recurrence of MSSA infection. We have added this to the revised manuscript.